# Vegetable Soups and Creams: Raw Materials, Processing, Health Benefits, and Innovation Trends

**DOI:** 10.3390/plants9121769

**Published:** 2020-12-14

**Authors:** Juana Fernández-López, Carmen Botella-Martínez, Casilda Navarro-Rodríguez de Vera, María Estrella Sayas-Barberá, Manuel Viuda-Martos, Elena Sánchez-Zapata, José Angel Pérez-Álvarez

**Affiliations:** 1IPOA Research Group, Agro-Food Technology Department, Higher Polytechnic School of Orihuela, Miguel Hernández University, Orihuela, 03312 Alicante, Spain; j.fernandez@umh.es (J.F.-L.); c.botella@umh.es (C.B.-M.); casilda.navarro@umh.es (C.N.-R.d.V.); estrella.sayas@umh.es (M.E.S.-B.); mviuda@umh.es (M.V.-M.); 2Research & Development Pre-Cooked Convenience Food, Surinver El Grupo S.Coop, 03191 Alicante, Spain; ejsanchez@surinver.es

**Keywords:** vegetable, soups, bioactive compounds, convenience foods, healthy foods, innovation

## Abstract

Vegetable soups and creams have gained popularity among consumers worldwide due to the wide variety of raw materials (vegetable fruits, tubers, bulbs, leafy vegetables, and legumes) that can be used in their formulation which has been recognized as a healthy source of nutrients (mainly proteins, dietary fiber, other carbohydrates, vitamins, and minerals) and bioactive compounds that could help maintain the body’s health and wellbeing. In addition, they are cheap and easy to preserve and prepare at home, ready to eat, so in consequence they are very useful in the modern life rhythms that modify the habits of current consumption and that reclaim foods elaborated with natural ingredients, ecologic, vegans, less invasive production processes, agroindustry coproducts valorization, and exploring new flavors and textures. This review focuses on the nutritional and healthy properties of vegetable soups and creams (depending on the raw materials used in their production) highlighting their content in bioactive compounds and their antioxidant properties. Apart from the effect that some processing steps could have on these compounds, innovation trends for the development of healthier soups and creams adapted to specific consumer requirements have also been explored.

## 1. Introduction

A soup is a flavorful and nutritious liquid food usually served at the beginning of a meal or a snack. Each soup should reflect its own identity. The flavor of the main ingredient used should remain prominent. Soups are broadly classified into two types—thick soups and thin (clear) soups. This is done based on the texture of the soups [1]. Thick soups are classified depending upon the type of thickening agent used: purées are vegetable soups thickened with starch; bisques are made from puréed shellfish or vegetables thickened with cream; cream soups may be thickened with béchamel sauce; and veloutés are thickened with eggs, butter, and cream. Other ingredients commonly used to thicken soups include rice, lentils, flour, and grains [2]. In the case of vegetable soups only ingredients from vegetable origin are allow.

Commercial soup became popular with the invention of canning in the 19th century, and today a great variety of soups are on the market. Canned soup can be condensed, in which case it is prepared by adding water or it can be “ready-to-eat”, meaning that no additional liquid is needed before eating. Microwaveable bowls have expanded the “ready-to-eat” canned soup market even more, offering convenience. Other way for soups commercialization is as dry soup mixes which need to be reconstituted with hot water before eating and other fresh ingredients may then be added to increase their flavor [3]. Dried soup powders have an advantage of protection from enzymatic and oxidative spoilage and flavor stability at room temperature over long periods of time (6–12 months). In addition, they are ready for reconstitution in a short time for working families, hotels, hospitals, restaurants, and institutional use as well as to military rations. Moreover, they exert light weight for shipping and availability at all time of the year [4,5,6,7].

Soup production is a rapidly developing and innovative category worldwide, with an already wide and growing range of value-added soup products with homemade taste and ready to-eat convenience. To analyze the economic importance of this sector it must be considered that, in general it is included in the “processed fruit and vegetables” food category.

According to data from a statistical overview of Eurostat, processed fruit and vegetables were worth € 47 billion, or 6.7% of the overall value of the EU food industry’s output and processing is concentrated in five countries. In addition to being consumed directly and traded as raw commodities, fruits and vegetables are processed into a multitude of food products, which can be grouped into frozen and preserved fruits and vegetables (canned vegetables, jams, marmalades, and dried fruits) (71.7%), juices (19.5%), tomato ketchup (3.7%), prepared meals (3.6%), and drained fruits, homogenized vegetables, and fruits (1.5 %). About prepared meals and dishes based on vegetables (where vegetable creams are included) its production value was situated in € 1.680 million [8].

Fruit and vegetable processing took place mainly in five member states, which were responsible for over two thirds (67.4%) of the total value of production. The leading member states were Italy (18.9%) and Spain (14.5%), which alone produced over one third of the total value. They were followed by Germany (11.8%), France (11.35%) and the United Kingdom (11.0%).

The EU was a net importer of processed fruit and vegetables: imports represented 18.5% of the value of its own output, while the value of exports represented 7.1% of the value of production. However, some countries registered values for exports higher than those for imports: among the big five producers, Spain and Italy recorded a trade surplus, while for France, Germany, and the United Kingdom, the trade balance was negative.

In Spain, according to data from MAPA [9], the consumption of prepared dishes (food category in which soups and vegetable creams are included) increased by 4.2% in 2019, compared to 2018 data. The average price of this type of product shows some stability (0.6%), with the average price of these products from € 4.30/kg. These two indicators cause that the category in terms of value obtain an increase of 4.9% compared to 2018. These products are increasingly in demand by Spanish individuals, in fact, the per capita consumption is 15.17 kg/person/year which is an amount 3.1% higher than ingested in the previous year. Per capita spending, therefore, also increases during the 2019, each Spanish individual spent 3.7% more on these products than in the period previous being the amount invested of 65.24. The category represents 4.33% of the total expenditure on food and beverages for the home. The purchase of prepared dishes of Spanish households since 2008 has been increasing. If we consider a medium-term dimension and compare since 2013, these food products have grown by 26.1% at the total category level, the trend is growing, especially since 2016–2017.

The types or segments of prepared dishes that have a growth equal to or greater than that of the category are ready meals canned, soups and creams, pasta ready meals, refrigerated tortillas and the rest of the prepared dishes. Taking into account what is the proportion by type of dishes prepared within the home in volume and value during the year 2019, soups and creams are the preferred prepared dishes for Spanish households, since they have a higher proportion in volume within Spanish homes. During 2019, its proportion in volume was 36.6% of the total kg of this category. In value, however, its share is much lower (12.8%). Its variation in both volume and value is positive with increases of 5.9% and 9.6%, respectively, growing above the category and as such being a growth engine for it [9].

In general, packaged soup is a flat category also in United States. Soup sales grew 1% from 2014 to 2019, and through 2024, when adjusted for inflation, the category’s total sales will be 5% shy of 2019′s performance [10].

The concern about the relationship between diet and health, the growing demand for healthy ready-to-eat products (according to current social habits), the boom in diets based on veganism and the possibilities that vegetable soups offer to innovate and develop new products (ingredients, tastes, process, packaging, etc.) are some of the reasons that have contributed to the current relevance of this type of foods. Considering that, in this review, we summarize and discuss recent advances about the composition and identification of bioactive compounds with healthy properties in the most common raw materials used in soup and creams formulation, the effect of processing on the main components of these raw materials and the innovation trends in the vegetable soups and creams sector. It is also aimed at exploring the health benefits associated to the consumption of vegetable soups and creams.

## 2. Main Raw Materials Used in Vegetable Soups and Cream Production

Different fruits vegetables, leafy vegetables, tubers, bulbs, legumes, and other herbal extracts have been used as main ingredients in soup formulation. Most of them are valuable sources of fiber, vitamins, and minerals and contain a large group of compounds that could help maintain the body’s health and well-being [11,12,13]. These are plant secondary metabolites, which, when consumed with a proper diet, can reduce the risk of age-related chronic diseases [14,15,16,17,18,19]. These secondary metabolites include phenolic acids, flavonoids, and glucosinolate derivatives, including isothiocyanates, terpenes, and low-molecular weight sulfur compounds, and carotenes, among others [20].

### 2.1. Fruit Vegetables

The most important fruits vegetables used in soups formulation are peppers, eggplants, pumpkins, and tomatoes.

#### 2.1.1. Pepper (*Capsicum* spp.)

Pepper is a fruit from the Solanaceae plant family, very popular in part of the world due to their color and aroma attributes [21]. The color of the peppers varies from green to black; in fact, exist yellow, orange, and red pepper too. The color of peppers depends on the maturity stage: green peppers are harvested before reaching maturity and red peppers are the most mature and have sweetest taste [22]. There are many species of peppers, however only five are commonly used: *Capsicum. annuum; Capsicum. baccatum; Capsicum. chinese; Capsicum. frutescens* and *Capsicum. pubescens* [23]. In general, peppers have a significant contribution to human nutrition due to its carotenes, polyphenols, antioxidant properties, fatty acids (including palmitic (16:0), oleic (18:1n-9), and linoleic (18:2n-6) acids), vitamins (C and E) and mineral substances. The main phenolics compounds found in peppers pulp are ferulic acid (404-2661 µg/kg of fresh weight (fw)), ellagic acid (896–2434 µg/kg fw), synaptic acid (181–2025 µg/kg fw), naringenin (73–968 µg/kg fw), vanillic acid (45–149 µg/kg fw), caffeic acid (38–63 µg/kg fw), 4-hydroxybenzoic acid (18–37 µg/kg fw), and p-coumaric acid (14–69 µg/kg fw) [24]. Other studies show that most flavonoids found in peppers are aglycones and glycosides of myricetin, quercetin, luteolin, apigenin, and kaempferol [23]. Carotenoids can act as antioxidants and can influence the flavor of the peppers. Total carotenoids level in pepper is between 59.86 and 1349.97 µg/g [25]. The carotenoids found in peppers are α-carotene (0.020 mg/100 g fw), β-carotene (0.5–1.64 mg/100 g fw), β-cryptoxanthin (0.5–1.2 mg/100 g fw), and non-provitamin A carotenoids (lutein + zeaxanthin) with values between 6 to 51 mg/100 g fw [26]. In fact, the orange-colored pepper is a source of zeaxanthin. Lutein, however, is the most abundant carotenoid in yellow peppers [23]. Ascorbic acid is another important bioactive in peppers species and some varieties of pepper contain about twice as much vitamin C as orange. Vitamin C in peppers reported in literature varied between 127 to 327.29 mg of ascorbic acid/100 g dw (dry weight). Peppers contain other vitamins as B6, A, K, and E. In fact, α and γ-tocopherol are found in large quantities in pepper, there are located in the tissue of pericarp and in seeds, respectively [23,26]. The most common minerals found in peppers, ordered from a lower to higher content are selenium, copper, boron, manganese, zinc, iron, sodium, calcium, magnesium, phosphorus, and potassium [26].

#### 2.1.2. Eggplant (*Solanum melongena* L.)

Eggplant is native to Asia, particularly to China, India, and Thailand. Eggplant is on the most widespread vegetable in the world. It is an important vegetable crop, due to its reported nutritional and economic values. Eggplants contain minerals (manganese, phosphorous, potassium and magnesium), vitamins (vitamin C, 2.20 mg/100 g fw; vitamin E, 0.30 mg/100 g fw; vitamin B1, 0,039 mg/100 g fw; vitamin B2, 0.037 mg/100 g fw; and vitamin B3, 0.649 mg/100 g fw) [27] and are classified as one of the richest sources of dietary fiber [28,29]. In addition, they are referenced as a potential source of valuable bioactive compounds [30]. There is a great variety of eggplants on the market with different shapes and color. The most consumed are purple eggplants that owe their color to anthocyanins. Eggplants show a high antioxidant activity which has been attributed to their high levels of phenolic compounds. The main family of phytochemicals found in the eggplant fresh is phenolic acids and anthocyanins which are mainly located in the eggplant peels [31,32]. The main phenolic acid of eggplant is chlorogenic acid (3-O-caffeoylquinic acid), representing from 70 to 95% of total phytochemicals in pulp of fresh eggplant (730–848 mg/ kg dry weight). It also has other phenolic compounds such as kaempferol, luteolin, tamarixetin, quercetin, baicalein, isorhamnetin, apigenin, caffeic acid, and ferulic acid [33]. Eggplant anthocyanins are represented by nasunine (delphinidin-3-(p-coumaroylrutinoside)-5-glucoside) and delphinidin-3-rutinoside as the major delphinidin-glycosides in eggplant peels [30]. The total phenol content, flavonoid content and antioxidant activity in eggplants depend on their color: white varieties showed a total phenol content of 40.02 mg GAE (gallic acid equivalent)/100 g, total flavonoid content of 22.55 mg CE (catechin equivalent/100 g, and an antioxidant activity of 2.91 g trolox /g sample measured by DPPH method; green varieties showed a total phenolic content of 26.10mg GAE/100 g, a total flavonoid content of 11.60 mg CE/100 g, and an antioxidant activity of 1.63 g trolox /g of sample; purple varieties showed a total phenol content of 24.31mg GAE/100 g, a total flavonoid content of 10.27 mg CE/100 g, and 0.9 g trolox /g of sample as antioxidant activity [34]. The total monomeric anthocyanins in purple genotype peels varied from 27.63 to 359.28 µg Cyanidin-3-glucoside/100 g fw and the antioxidant activity measured by the DPPH method was 66.74 ± 4.60 µg / mL and measured by the ABTS method of 53.18 ± 0.71 µg / mL [35].

#### 2.1.3. Tomato (*Solanum lycopersicum*)

Tomato is the most produced vegetable in the world. It is consumed both fresh and processed as canned tomato, tomato puree, tomato paste, tomato juice and included in multitude of dishes and sauces [36]. Tomatoes are rich in potassium (204–248 mg/100 g), sodium (13–42 mg/100 g), calcium (5–13 mg/100 g), magnesium (8–12 mg/100 g), phosphorous (28–36 mg/100 g), vitamin A (90 μg eq. retinol/100 g), vitamin C (9–23.4 mg/100 g), and vitamin E (0.38 mg eq. tocopherol). In addition, tomato has important bioactive compounds and antioxidants components as polyphenolics compounds, carotenoids, folates, ascorbic acid, and vitamin E [37]. Phenolic acids are the most representative phenolic compounds found in tomato and processed products, including hydroxybenzoic and hydroxycinnamic acids and their ester conjugates, with chlorogenic acid being one of the most abundant [38]. A high content of the chalcone and flavanone forms of naringenin and the flavonol rutin has also been described in tomato-based products [39]. The concentration of bioactive compounds shows great variability due to the many factors that affect their biosynthesis in the plant. Lycopene content varies throughout maturation [40]. The phenolics compounds analyzed by Gómez-Romero et al. [41] in extracts from three tomatoes varieties of cultivars from Almería, in fact Raf, Rambo and Daniela variety showed 7.34–36.84 mg of hydroxybenzoic acids and derivatives/100 g dw; 149.36–197.65 mg of hydroxycinnamic acids and derivates/100 g dw, 14.99–77.33 mg of phenylacetic acids and derivatives/100 g of dw, and 183.87-343.00 mg of flavonoids and glycosides/100 g of dw.

#### 2.1.4. Pumpkin (*Cucurbita* spp.)

Pumpkins include different varieties with specific colors (green, yellow, and orange), taste and texture. The two varieties more used for soup formulation are the common or Halloween pumpkin (*Cucurbita maxima*) due to their big amount of pulp, and the butternut (*Cucurbita moschata*) due to its orange and creamy flesh and sweet flavor. Pumpkins contain minerals as calcium (21 mg/100 g fresh weight), phosphorous (44 mg/100 g fw), iron (0.8 mg/100 g fw), sodium (1 mg/100 g fw), and potassium (340 mg/100 g fw). In addition, they contain vitamins (vitamin A, 1600 IU; thiamin 0.05 mg/100 g fw; riboflavin 0.11 mg/100 g fw; niacin 0.60 mg/100 g fw; ascorbic acid and 9.0 mg/100 g fw). The dietary fiber content of pumpkins is 1.1 g/100 g fw for two compact tropical pumpkin hybrids (*Cucurbita moschata)* [42]. In the study carried out by Kulczyński and Gramza-Michałowska [43] where 11 pumpkin cultivars of the *Cucurbita maxima Duchesne* spp were analyzed for the content in minerals and vitamins, they reported 4692.70–9965.70 mg/100 g dw for potassium, 92.12–264.89 mg/100 g dw for calcium, 79.97–135.35 mg/100 g dw of magnesium, 226.16–370.40 mg/100 g dw of sodium, 1.06–2.67 mg/100 g dw of iron, 0.79–1.33 mg/100 g dw of zinc, 0.25–0.59 mg/100 g dw of copper, and 0.34–0.90 mg/100 g dw of manganese. The vitamins mainly are vitamin C (49.06–84.23 mg/100 g dw), vitamin B1 (0.15–0.60 mg/100 g dw), and folates (20.44–65.04 µg/100 g dw).

Pumpkins are reference as a potential source of bioactive compounds like as polyphenols, carotenoids, tocopherols, sterols, bioactive proteins, peptides, dietary fiber, pre- and probiotics and fatty acids. There is a great variation of this type of compounds according to the cultivar analyzed, so the main phenolic acid found in 11 pumpkin cultivars of the *C. maxima Duchesne* species and in 15 pumpkin varieties belonging to the *C. pepo* and *C. moschata* are caffeic acid (8-46–133.42 mg/100 g fw), protocatechuic acid (4.50–52.55 mg/100 g fw), gallic acid (2.58–25.62 mg/100 g fw), 4-hydroxy-benzoic acid (4.03–33.12 mg/100 g fw), vanillic acid (0.48–9.49 mg/100 g fw), chlorogenic acid (1.23–8.00 mg/100 g fw), p-coumaric acid (0.04–3.47 mg/100 g fw), ferulic acid (1.90–41.41 mg/100 g fw), and sinapic acid (7.24–32.00 mg/100 g fw) [43,44]. These studies showed that the main flavonoids found in pumpkins are rutin (2.02–51.92 mg/100 g fw), kaempferol (1.06–36.24 mg/100 g fw), isoquercetin (1.00–9.01 mg/100 g fw), astragalin (2.68–28.03 mg/100 g fw), myrcetin (0.89–9.04 mg/100 g fw), and quercetin (1.92–26.54 mg/100 g fw). Total carotenoids content in the pumpkin pulp are between 10.8 and 57 mg/ kg fw for several pumpkin hybrids from *Cucurbita moschata* [42,45]. The carotenoids found in pumpkins are zeaxanthin (0.31–192.53 µg/100 g fw), lutein (3.34–388.79 µg/100 g fw), β-carotene (1.29–115.29 µg/100 g fw), and retinol equivalent (0.52–67.66 µg/100 g fw). Other compounds such as tocopherols are present in pumpkins in particular α-tocopherol, γ-tocopherol, and α-tocopherol equivalent [43,44]. The total phenol content, flavonoid content and antioxidant activity in pumpkins depend on cultivar variety and solvents for extraction. The total phenol content from *C. maxima* of pulp and peel show 13.92–33.48 mg GAE/100 g fw (water or 50:50 acetone–ethanol) and 496.97 mg GAE/100 g dw for the same variety, total flavonoid content of 3.79–11.72 mg quercetin equivalent/100 g fw (water or 50:50 acetone–ethanol) [46,47]. The antioxidant activity of 71.1–387 µg trolox equivalent (TE)/g sample measure by DPPH method (Priori et al., 2106); 78.21 µM TE/g dw of sample measure by ABTS method and 69.37 µM Fe_2_SO_4_/g of dw sample, measure by FRAP method [46].

### 2.2. Green Leafy Vegetables

They are considered an excellent source of dietary fiber and also have a high content in chlorophylls (responsible for their green color), carotenoids, vitamins C and K, calcium, magnesium, iron, potassium, and folic acid. There is a great variety of green leafy vegetables but the most used in soup formulation are chard and spinach.

#### 2.2.1. Chards (*Beta vulgaris* var. *cicla*)

Chards are characterized by their big brilliant green leaves. Nutritionally it is rich in dietary fiber (2 g/100 g fw), vitamins (vitamin K, 327 µg/100 g fw; vitamin A, 306 µg/100 g; vitamin C, 18 mg/100 g fw), minerals (magnesium, 86 mg/100 g fw; potassium, 549 mg/100 g fw; iron, 2.26 mg/100 mg fw). The main phenolics compounds found in chards are hydroxycinnamic acids as p-coumaric and rosmarinic and myricitrin acids and their derivates. Myricitrin acid are the major identified (4.08 mg/ g extract) follow by p-coumaric (3.53 mg/g extract) and rosmarinic acid (1.02 mg/g extract) from leaves of Swiss chard (*Beta vulgaris L.* var. *cicla*) collected in Tunisia [48]. Pyo et al. [49] showed a difference between red and white varieties of chard and different amount of phenolic acids in leaf or stem. The among of syringic acid (45.1–1.5 = mg/100 g fw), caffeic acid (14.8–0.9 mg/100 g fw), p-coumaric acid (10.4– mg/100 g fw), ferulic acid (10.8–3.4 mg/100 g fw), chlorogenic acid (7.7–0.8 mg/100 g fw), protocatechuic (5.4–1.7 mg/100 g fw), vanillic acid (5.4–0.9 mg/100 g fw), p-OH-benzoic acid (4.7–1.9 mg/100 g fw), and gallic acid (3.7–1.2 mg/100 g fw). The amount of total phenol compounds is in the range of 128–101.5 mg/100 g fw. In general, the leaves contain more phenolic acids than the stems, and between the two varieties the quantity of these phenolic acids varies. The study of Maucieri et al. [50] reported caffeic acid (2.1–10.09 mg/ kg dw), p-coumaric acid (257–400 mg/ kg dw), ferulic acid (24.7–53.4 mg/ kg dw), and chlorogenic acid (200–0357 mg/ kg dw) for white leaves of Swiss chard. Other studies showed that the main flavonoids found in chards are catechin (6.7 mg/100 g fw), myricetin (2.2 mg/100 g fw), quercetin (7.5 mg/100 g fw), and kaempfenol (9.2 mg/100 g fw) [49], and flavonoid glycosides derivate from apigenin (vitexin, vitexin-2-O-rhamnoside, vitexin-2-O-xyloside [51]. The color for red varieties is due to anthocyanins and their content is 0.47 μM/g for chard extract [52]. All these compounds (phenolic acids, flavonoids, and anthocyanins) contribute to the antioxidant activity of chard; thus, Pyo et al. [49] analyzed the antioxidant activities of methanol extracts obtained from leaves, as well as stems of white and red tissue cultivars using the DPPH and thiocyanate assays. These authors reported that using the DPPH radical scavenging method and the thiocyanate method, the antioxidant activity of each extract from Swiss chard was in the following order: red leaf > white leaf > red stem > white stem. This result was consistent with the amount of total phenolic content of each extract (red leaf 128.1 mg/100 g fw, white leaf 101.5 mg/100 g fw, red stem 29.7 mg/100 g fw, and white stem 23.2/100 g f.)

#### 2.2.2. Spinach (*Spinacia oleracea*)

It is native from Persia and it is characterized by its dark green leaves. It is the vegetable that is most frequently frozen. Nutritionally is a good source of dietary fiber, vitamins (vitamin A, 469 µg/100 g fw; vitamin E (2 mg/100 g fw; vitamin K, 483 µg/100 g fw; vitamin B9, 194 µg/100 g fw) and minerals (magnesium, 79 mg/100 g fw; potassium, 558 mg/100 g, calcium, 99 mg/100 g fw). In their composition it is possible to find several bioactive compounds mainly phenolic or flavonoid such as quercetin, patuletin, spinacetin, and jaceidin, ferulic acid, vanillin, and caffeic acid [53]. Other structures mentioned in the literature are flavones and derivatives of patuletin, spinatoside, spinacetin, jaceidin, as well as lignans such as lariciresinol, secoisolariciresinol, and pinoresinol [54]. Organosulfur compounds such as S-allylcysteine and S-methylcysteine, also are found in spinach [55]. As mentioned above, spinach is rich in antioxidant components. Antioxidant capacity of leaves and petioles of spinach with different leaf types was analyzed by Yosefi et al. [56]. These authors reported that the antioxidant capacity, measured with the ABTS assay, ranged from 25.62 to 64.06 μmol trolox/g dry weight for leaves and from 12.70 to 20.34 μmol trolox/g dw for petioles. In a similar study, Ko et al. [57] analyzed the antioxidant activity of two extracts (water and ethanol extracts) obtained from spinach using the oxygen radical absorbance capacity (ORAC) assay. They reported that the ORAC values of spinach water extract and spinach ethanol extracts increased in a concentration dependent manner. At analyzed concentration of 50 μg/mL, water extract and ethanol extracts showed similar ability to protect a fluorescent reporter from oxidative degeneration by AAPH (7.6 and 7.2 TE, respectively).

### 2.3. Tubers

Tubers are nutritionally characterized by being rich in carbohydrates (mainly starch), dietary fiber, a small amount of protein, some minerals (mainly potassium) and few vitamins. Due to their richness in starch they are used in the elaboration of soups because they give consistency to the final product, and also in some cases for their attractive color (mainly orange color). Within this group, carrot, potato, and sweet potato have been revealed as the most suitable for their application in soup formulation.

#### 2.3.1. Potato (*Solanum tuberosum*)

Potato is originally from the Andean highlands (Peru and Bolivia) and nowadays it is one of the most important crops in the world for human consumption. It has a high nutritional value and its easy digestibility made it an excellent ingredient for soup formulation. It is composed mainly by starch (18% fw), some vitamins (vitamin C, 20 mg/100 g fw; vitamin B6, 0.25 mg/100 g fw), and minerals (potassium, 421 mg/100 g; phosphorous 57 mg/100 g). In addition to high quality proteins, potato tubers amass significant quantities of vitamins and minerals, as well as an assortment of phytochemicals including phenolics, phytoalexins, and protease inhibitors. Potatoes are good sources of phenolic acids, with total phenolic amount greater than other vegetables such as carrots, onions, and tomatoes. Chlorogenic acid constitutes up to 90% of the potato tuber natural phenols. Others found in potatoes are 4-O-caffeoylquinic acid (cryptochlorogenic acid), 5-O-caffeoylquinic (neo-chlorogenic acid), 3,4-dicaffeoylquinic, and 3,5-dicaffeoylquinic acids [58]. Other phenolic acids such as caffeic acid, ferulic acid, gallic acid, and p-coumaric acid have also been quantified in potatoes, ranging from 0 to 5 mg/100 g dry weight [59]. In these tubers, also is possible to find several flavonoids. The most abundant is catechin, with values ranged between 0 and 204 mg/100 g dry weight [60]. Other flavonoids such as quercetin, rutin, and kaempferol rutinose are also present in potato tubers [61]. In purple and red potatoes varieties also is possible to find anthocyanins. Anthocyanins that are responsible for blue to purple coloration of potato tuber tissues are mainly petunidin 3-p-coumaroylrutinoside-5-glucoside, petunidin 3-feruloylrutinoside-5-glucoside, and malvidin 3-p-coumaroylrutinoside-5-glucoside. Responsible for the red coloration are mainly pelargonidin 3-feruloylrutinoside-5-glucoside, pelargonidin 3-rutinoside, pelargonidin 3-rutinoside-5-glucoside, and pelargonidin 3-p-coumaroylrutinoside-5-glucoside [62]. Others important bioactive compounds found in potatoes are the carotenoids. Thus, higher levels of total carotenoids were found in the skin and flesh of yellow cultivars being all-trans-Lutein, all-trans-antheraxantin, all-trans-zeaxanthin and all-trans-β-carotene the principal compounds with values ranging between 0.55 and 12 mg/kg [63,64]. Due to the content in bioactive compounds, potato showed a moderate antioxidant activity. Therefore, Hamouz et al. [65] analyzed the antioxidant activity of potato flesh with different colors (four yellow- or white-, six purple-, and four red-fleshed varieties) grown in the Czech Republic. These authors reported that in average values, the lowest antioxidant values were found in the group of yellow- or white-fleshed varieties (ascorbic acid equivalent 82.83 mg/kg), while in the group of red-fleshed varieties antioxidant activity was found as 359.38 mg ascorbic acid equivalent /kg of fw, i.e., higher 4.34 times, and in the group of purple-fleshed varieties even 5.03 times higher (416.54 mg ascorbic acid equivalent/kg fw).

#### 2.3.2. Sweet Potato (*Ipomea batatas* L.)

Sweet potato is originally from America and is characterized by its sweet flavor due to a high content in simple carbohydrates (sucrose, glucose, and fructose). Depending on the variety, their color can vary from white to yellow or orange. The higher the orange color, the higher the amount of carotenes. It has an energetic value higher than that of potatoes and stands out for its content in provitamin A (β–carotene). Orange, yellow, purple, and white sweet potato varieties may differ not only on their skin or flesh color but also on their nutritional composition and profile of bioactive compounds. As regards to phenolics acids, Sasaki et al. [66] reported that the main phenolic acid found in purple sweet potato are 5-caffeoylquinic acids: 3,4-, 4,5-, and 3,5-dicaffeoylquinic acids with values ranged between 10.7 and 53.3 mg/100 g. Similarly, Grace et al. [67] mentioned that the principal phenolic compounds present in purple, orange, light yellow, and yellow sweet potato were chlorogenic acid, caffeic acid, 3,4-, 4,5-, and 3,5-dicaffeoylquinic acids with values comprised between 0.34 and 16.76 mg/g. Another important bioactive compound present in sweet potato are the anthocyanins. Lim et al. [68] and Esatbeyoglu et al. [69] mentioned that the principal anthocyanins found in purple sweet potato were peonidin-3-sophoroside-5-glucose, cyanidin-3-p-hydroxybenzoylsoph-5-glucose, cyanidin-3-(6′″caffeoylsoph)-5-glucose, peonidin-derivative, peonidin-3-p-hydroxybenzoylsoph-5-glucose, cyanidin-3-feruloylsoph-5-glucose, peonidin-3-feruloylsoph-5- glucose, cyanidin-3-(6″caffeoylsoph)-5-glucose, cyanidin-3-(6″,6′″-dicaffeoylsoph)-5-glucose, cyanidin-3-(6″-caffeoyl-6′″-p-hydroxybenzoylsoph)-5-glucose, peonidin-3-(6″caffeoylsoph)-5-glucose, peonidin-3-(6″,6′″-dicaffeoylsoph)-5-glucose, peonidin-3-(6″-caffeoyl-6′″-p-hydroxybenzoylsoph)-5-glucose, and peonidin-3-(6″-caffeoyl-6′″-feruloylsoph)-5-glucose, with values range between 2.4 and 68.4 mg/100 g. Another important phytochemical found in sweet potato are the carotenoids. Thus, Grace et al. [67] reported that β-carotene is presented in purple, orange, light yellow, and yellow sweet potato varying from 1 to 253 μg/g. Ishiguro et al. [70] analyzed the total content and composition of carotenoids in yellow and orange sweet potatoes. In yellow cultivars the carotenoid content ranged from 1.131 to 3.908 mg/100 g dw whilst for orange cultivars the carotenoid content varied from 14.791 to 46.187 mg/100 g dw in orange-fleshed cultivars. The main carotenoids were β-carotene-5,8,5′,8′-diepoxide (approximately 32%–51%) and β-cryptoxanthin-5,8-epoxide (approximately 11%–30%) in the yellow, whereas β-carotene (approximately 80–92%) was dominant in orange cultivars. Due to the high content in bioactive compounds sweet potato shows a considerable antioxidant capacity. In this way, Teow et al. [71] analyzed the antioxidant activities (μmol TE/g fw) in 19 sweet potato genotypes with distinctive flesh color (white, cream, yellow, orange, and purple) using ORAC assay. These authors reported that the total antioxidant activity (hydrophilic + lipophilic ORAC) was highest (27.2 μmol TE/g fw) for purple cultivars and lowest (2.72 μmol TE/g fw) for white sweet potatoes. Ji et al. [72] analyzed the antioxidant activity of four different color fleshed sweet potatoes, purple, red, yellow, and white. They reported that the antioxidant capacity of extracts was 81.2 mg/g dw for purples sweet potato which had the highest antioxidant capacity values followed by white (55.2 mg/g dw) and red (50.4 mg/g dw) while yellow sweet potato (43.3 mg/g dw) was the lowest values.

#### 2.3.3. Carrots (*Daucus carota*)

Carrot is the second popular vegetable after potatoes [73]. Carrots contain several compounds that can be used in the food and chemical industries: sugars (9–11%), carotenoids (0.01–0.02%), pectins (1–1.5%), and other fibers (2–5%). Main sugars present are sucrose, fructose and glucose [73]. In their composition, it is possible to find several phytochemicals mainly carotenoids and in some cases anthocyanins. Therefore, in a study carried out by Nicolle et al. [74] the carotenoid content in some carrot cultivars was analyzed, they found that carotenoid content varied deeply between cultivars, including white, yellow, orange, and purple. Their study found that white carrots contain 0.32 mg/100 g fw, whereas dark orange carrots contain approximately 17 mg/100 g fw. β-carotene represents 65% of the total carotenoid content in the orange carrot. In other study Dias [75] reported that lutein represents almost 50% of the total carotenoid content in yellow and purple carrots. As regards to anthocyanins content, some studies have reported the presence of cyanidin-3-lathyroside, cyanidin-3-β-D-glucopyranoside, and different glucosides of cyanidin, malvidin, and peonidin [76,77]. Furthermore, it is possible to find several phenolic acids including caffeoylquinic acid; 5′-caffeoylquinic acid; caffeic acid; cis-5′-caffeoylquinic acid; 4′p-coumaroylquinic acid; 3′4′-dicafferoylquinic acid; and 3′5′-dicafferoylquinic acid [78]. The antioxidant properties of carrots are widely demonstrated. Therefore, Dong et al. [79] determined the in vitro antioxidant activity of carrot using different kinds of assays, including DPPH and ORAC capacity. These authors reported concentrations between 50 and 500 μg/mL, the scavenging ability of carrot on DPPH radicals ranged from 28.8 to 82.2% (and with EC50 values of 157.58 μg/mL). As regards ORAC assays, the relative ORAC value of carrot increased from 2.923 μM trolox to 12.111 μM trolox when the concentrations increased from 1 to 10 μg/mL. Algarra et al. [80] analyzed the antioxidant capacity of two different black carrots cultivars. The authors mentioned that the reducing capacity of the two black carrots extracts were 86.4 and 182.0 μM TE /100 g fw while the radical scavenging ability was 17.6 and 240.0 trolox equivalents/100 g.

### 2.4. Bulbs

The main bulbs that are consumed across the world are from *Allium* genus. *Allium* is the largest and most important representative genus of the Alliaceae family and comprises 450 species, widely distributed in the northern hemisphere. In addition to the well-known garlic (*Allium sativum*) and onion (*Allium cepa*), several other species are widely grown for culinary use, such as leek (*Allium porrum L.*), scallion (*Allium fistulosum L.*), shallot (*Allium ascalonicum Hort.*), wild garlic (*Allium ursinum L.*), elephant garlic (*Allium ampeloprasum L.* var. *ampeloprasum*), chive (*Allium schoenoprasum L.*), and Chinese chive (*Allium tuberosum L.*) [81]. Most of them are used in soup formulation to enhance their flavor and texture.

#### 2.4.1. Onions (*Allium cepa* L.)

Onions are the most used flavoring vegetable in the world. There are a lot of varieties with different colors (brown, white, and red), shapes, size, texture, and intensity of flavor. Onions are not rich in the common nutrients but are low in energy. The main minerals in onions are potassium (12,720.67–13,550.1 µg/g), sulfur (3415.67–3421.58 µg/g), calcium 2506.5–3183.54 µg/g), phosphorus (2525.63–2677.64 µg/g), magnesium (980.43–1100.62 µg/g), sodium (314.13–1001.34 µg/g), iron (27.69–23.16 µg/g), boron (15.34–18.34 µg/g), zinc (12.23–14.43 µg/g), Manganese (6.03–9.22 µg/g), and copper (3.19–7.38 µg/g) [82]. They are rich in bioactive compounds—flavonoids, fructans, saponins, and sulphur containing compounds. The red varieties are a source of the flavonoids and anthocyanins. Onions are a good source of flavonoid compounds; in particular, quercetin and quercetin derivatives that account for up to 93% of the total flavonol content [83]. Another 25 different flavonols have been identified in onions, such as kaempferol, isorhamnetin, and myricetin with their derivates [84]. The quantity of these flavonols in onions is around 250 mg/kg of which 0.3 mg/g fw is quercetin [85]. Onion skins contain significantly higher levels of flavonoids (2–10 g/kg) [86]. Anthocyanins of onions are mainly presented in red varieties as glycosides of cyanidin, peonidin, and pelargonidin in proportion of 250 mg/kg [87]. The content of anthocyanins in some red onion has been reported to be approximately 10% of the total flavonoid content. Total phenol content fluctuates between 104.9 and 16.8 mg GAE/100 g for ten varieties of onions [88] and between 56 and 186 mg GAE/ mL of extract for white, yellow, and red onions in different extractants [89]. The total flavonoid compounds are between 69.2 and 5.8 mg of CE/100 g of sample [88] and for honeysuckle onion and sweet Italian onion are 345 and 449 µg/g fw, respectively, and the total anthocyanin for two red onions varieties are between 86 to 103 µg/g fw [82]. Several studies evaluated the antioxidant capacities for onions and found that the antioxidant activity for stored onion, onion and welsh onion were between 1.24 and 2.43 mg/ mL for IC50 by DPPH method and between 1.64 and 4.00 mg/ mL for IC50 by ABTS method [90]. Zill-e-Huma et al. [91] evaluated the antioxidant activity for four varieties of *Allium cepa* with DPPH and ORAC methods, so white onion and grelot showed higher inhibition (74.54–82.76 g/L for IC50 and 63.58–85.18 g/L for IC50, respectively) than red onion (17.09–18.00 g/L for IC50) by DPPH method. The antioxidant activity measured by the ORAC method was 29–201 µmol TE/g of sample.

#### 2.4.2. Leek (*Allium ampeloprasum* var. *porrum*)

Leek is a bulbous herb of the genus Allium. Leek is cultivated in Asia, America and Europe, especially in the Mediterranean region [92]. On the nutritional aspect, leeks are rich in vitamin C with a quantity of 18 mg/100 g fw, vitamin B6 (0.3 mg/100 g fw), 0.4 mg/100 g niacin, 96 µg/100 g of folate, and vitamin A and E with 83 and 0.73 µg/100 g, respectively. Regarding minerals, leeks contain calcium, iron, potassium magnesium, sodium, phosphorus, copper, manganese, and zinc [93]. Leeks contain a wide array of bioactive compounds like as organo-sulphur compounds which provide flavor, phenolic compounds, amino acids, and organic acids that may have promoting antimicrobial properties [94]. Among the phenolic compounds, leek is source of flavonoids like kaempferol derivates, quercetin derivates, phenolic, saponins, steroidal saponin, and essential oils [92]. The total phenolic acids compounds present in leaves and stem extracts of leeks ranged between 0.71 ± 0.12 for leaves and 2.44± 0.11 mg/g dw for stem, total flavonols between 2.44 ± 0.09 and 3.22± 0.12 mg/g dw, respectively, and total polyphenol content between 3.14± 0.11 mg/g dw and 5.66± 0.12 mg/g dw for leaves and stem, respectively [94]. The following phenolic compounds have been identified: gallic acid, protocatechuic acid, dihydroxybenzoic acid caffeic acid, vanillic acid, chlorogenic acid, ferulic acid, sinapenic acid, rosmarinic acid, syringic acid, quercetin, rutin, myricetin, kaempferol glucioside, quercetin glucoside, naringenin, and aspigenin [94,95]. Extracts of the white shaft and green leaves of 30 leek cultivars were investigated for their antioxidant properties and total phenolic content [96]. The total phenolic content ranged from 5.31 ± 0.75 to 13.96 ± 0.71 mg GAE/g dw. The antioxidant activity of leeks ranged from 2 to 14 µmol TE/g dw by DPPH method for three white bulbs and from 5 to 14 µM TE/g dw for green leaves. The FRAP results of the white bulb ranged from 5 to 28 µmol FeSO_4_/g dw and the results of the green leaves ranged from 17 to 100 µmol FeSO_4_/g dw [96]. Several studies showed a positive correlation between antioxidant power and the content of phenolic compounds in leeks [97,98].

#### 2.4.3. Garlic (*Allium sativum*)

Garlic is an annual bulbous herb native to Central and South Asia. The countries with the highest production are USA, China, India, Egypt, and South Korea [99]. Many studies have demonstrated that garlic has antifungal and antibacterial properties, which have been associated to the thiosulfinates and volatile sulphur compounds, which are responsible for the intense flavor of these vegetables. Nutritionally garlics contain polysaccharides, lipids, amino acids, proteins, vitamins and minerals (copper, 0.3–9.12 mg/kg of sample; magnesium, 0.77–1056 mg/kg of sample; and phosphorus, 140–6009 mg/kg of sample) [81,100,101]. Garlics are characterized by polar compounds of phenolics and steroidal origin as glycosylated. Moreover, garlic contained more than 20 phenolic compounds with higher contents than many common vegetables. The main phenolic compound was β-resorcylic acid, followed by pyrogallol, gallic acid, rutin, protocatechuic acid, as well as quercetin [102]. Total phenol content fluctuates between 0.05 and 19.4 mg GAE/g of fw. Total flavonoids compounds in fresh garlic are between 0.042 and 3.37 mg of CE/g of sample. Total flavanols and tannins content for fresh garlic are 17.45–67.05 µg CE/g of sample and 1.40–2.40 mg CE/g of sample, respectively [100,103,104]. Several studies evaluated the antioxidant capacities of raw garlic and found that the raw garlic exhibits a strong antioxidant activity by DPPH radical scavenging assay, ABTS radical scavenging assay, and FRAP assay. They reported values of 7–34.86 µM TE/g for the DPPH method; 6.90–10.80 µM TE/g for the FRAP; and 23.71–37.02 µM TE/g for ABTS method [101,103,105]. The major active components of garlics are their organosulfur compounds, such as allicin (diallyl thiosulfonate), DAS (diallyl sulfide), DADS (diallyl disulfide), DATS (diallyl trisulfide), E/Z-ajoene, S-allyl-cysteine (SAC), and alliin (S-allyl-cysteine) [106].

### 2.5. Legumes

Legumes have been an integral part of human nutrition for a very long time; they include several edible species such as lentils, beans, chickpeas, and peas. They are highly recommended by health organizations due to their composition, rich in inexpensive and sustainable proteins, slowly digestible carbohydrates, fiber, vitamins, minerals, and bioactive compounds [107,108].

#### 2.5.1. Lentil (*Lens culinaris*)

Lentil is one of the most popular legumes by millions in developed and developing countries due to the growing demand for healthy, constitutes a source of inexpensive proteins accessible to all. The major essential amino acids from lentils are arginine, leucine, and lysine, whereas the main non-essential ones are glutamic acid, aspartic acid, and serine [109]. Furthermore, lentils are considered to be an important source of phytochemicals, which makes lentils a functional food with several beneficial health effects (anticarcinogenic, blood pressure-lowering, hypocholesterolemic, and glycemic load lowering, anti-inflammatory, and antioxidants) [110]. Lentils exist as a spectrum of colors, which includes yellow, orange, red, green, brown or black, depending on the cultivar, the composition of the seed coats and cotyledons [111]. These pulses have the highest total phenolic content. Among polyphenols, tannins and tannin-related compounds are principal in lentils, and they are mainly concentrated in the testa. Recent reports indicate that catechin glucosides, procyanidin dimers, quercetin diglycoside, and trans-p-coumaric acid were the main phenolics in green lentils, while quercetin diglycoside, catechin, digallate procyanidin, and p-hydroxybenzoic were the dominant phenolics in red lentils [112,113]. As regard flavonoid content, some studies showed the high content in isoflavones like formononetin, daidzein, genistein, glycitein, and lignans, such as matairesinol, biochanin A, coumestrol, lariciresinol, pinoresinol, secoisolariciresinol, and coumestrol [114]. Further, it is possible to find others bioactive compounds like phytosterols. Thus, Ryan et al. [115] reported that the main phytosterols found in lentils are β-sitosterol, campesterol, and stigmasterol. The antioxidant activity of lentils has been widely determined. Therefore, Alshikh et al. [116] reported that trolox equivalent antioxidant capacity values of fractions containing esterified phenolics, which ranged from 0.14 to 1.49 mM TE/g of defatted sample, was higher than those of fractions containing free phenolics and insoluble-bound phenolic counterparts, with values in the range of 0.02–0.18 and 0.09–0.93 mM TE/g of defatted sample, respectively.

#### 2.5.2. Bean (*Phaseolus vulgaris* L.)

It is a legume widely consumed throughout the world and it is recognized as the major source of dietary protein in many Latin-American and African countries. They are a particularly useful all-round and are an excellent source of dietary fiber, a good source of folate, niacin, thiamin, and vitamin C, a source of riboflavin, vitamin A, vitamin B6, copper, iron, magnesium, and phosphorus and contain potassium at levels of dietary significance [117]. In addition, a high content of bioactive compounds are present in their composition. In purple, red and dark beans the principal compounds are flavonoids and anthocyanins including hesperetin 7-neohesperidoside, hesperetin 7-rutinoside, hesperetin 7-O-glucoside, hesperetin derivatives, naringenin 7-O-neohesperidoside, naringenin 7-O-rutinoside, naringenin 7-O-glucoside, naringenin-7-methyl ether 2, naringenin, quercetin 3-O-rutinoside, quercetin 3-O-galactoside, quercetin 3-O-glucoside, quercetin 3-O-glucoside acetate, quercetin, kaempferol 3-O-glucoside, kaempferol 3-O-rutinoside acylated, kaempferol, myricetin 3-glucoside, myricetin derivatives, myricetin, apigenin 7-O-glucoside, 7-O-glucoside, delphinidin 3-O-glucoside, cyanidin 3-O-glucoside, pelargonidin 3, 5-diglucoside, pelargonidin 3-O-glucoside, delphinidin glucoside acylated, pelargonidin glucoside acylated, pelargonidin 3-malonylglucoside, petunidin feruloyl glucose, petunidin derivatives, and malvidin derivatives [118,119]. Further, several phenolic acids also can be identified like gallic acid, protocatechuic acid, ferulyl aldaric acid, trans ferulic acid, p-coumaryl aldaric acid, trans p-coumaric acid, sinapyl aldaric acid, and sinapic acid [120].

#### 2.5.3. Chickpea (*Cicer aretianum* L.)

It could be considered an exceptional constituent in the human diet, due to providing proteins, fibers, starch, and several bioactive compounds [121]. The chickpea proximal composition include proteins (18.4–29.0%) which exhibit high solubility and digestibility (89.0%) and lipids (4.5–6.6%) with a fatty acid profile of palmitic (10.8%), oleic (33.5%), linoleic (49.7%), and linolenic (2.4%) acids [122,123]. As regards to bioactive compounds, the main carotenoids present in chickpea include β-carotene, lutein, zeaxanthin, β-cryptoxanthin, lycopene and α-carotene. The average concentration of carotenoids is higher in wild accessions of chickpea than in cultivated varieties or landraces (Abbo 2005). As occurs with the lentils also is possible to find isoflavones and lignans including biochanin A and formononetin daidzein, genistein matairesinol, and secoisolariciresinol [124]. Regarding to polyphenolic compounds, Quintero-Soto et al. [125] reported that sinapic acid hexoside, gallic acid, benzoic acid, dihydroxybenzoic acid, vanillin, p-hydroxybenzoic acid, p-coumaric acid, and ferulic acid hexoside were the main phenolic acids present in chickpea, while Aguilera et al. [126] reported that the principal flavonoids were pinocembrin, quercetin 3-O-rutinoside, kaempferol 3-O-rutinoside, 5,7-dimetoxyflavone, myrcetin, rutin, and isorhamnetin. The chickpea also shows antioxidant activity. Quintero-Soto et al. [125] analyzed the antioxidant activity of 18 chickpea genotypes using three different methodologies (ABTS, DPPH, and FRAP). These authors reported antioxidant activity values (μM TE/100 g dw) by ABTS (278-2417), DPPH (52-1650), and FRAP (41-1181).

Figure 1 shows the main bioactive compounds in the ingredients frequently used for vegetable soups formulation.

## 3. Processing Effects on the Main Components of Raw Materials

Industrial processing of vegetables purees, soups, and creams and the other requires of several preliminary operations (washing, peeling, and cutting), depending on the vegetables employed. Processing of vegetables soups is shown in Figure 2, for the two packing systems most used: aseptic and hot filling. Vegetables may be initially cleaned by dry methods to remove dust and coarse surface residues. They must be then washed with potable water by immersion or shower methods. Chlorine (100–200 ppm NaClO, pH 6.5–7.5) could be added to prevent cross-contamination in dump washing tanks. After removal some non-edible parts, depending on the product, they are then peeled, cut, or ground. After that, there are many food processing steps, such as blanching, steaming, boiling, (both, with or without pressure), vacuum, microwaving, stir-frying, pasteurization, canning, cooling, and freezing, and the majority have a strong impact on nutritional quality, as well as stability of bioactive compounds presented in vegetables [127]. As it is well known, treatment that involves the increase of temperature causes quantity and quality changes of phytochemicals, that may undergo oxidative degradation or be leached into the water during home cooking and industrial processing [128,129].

Vegetables’ blanching refers to the unit operation where vegetables are scalded in boiling water or steam before processing to inactivate enzymes. Special precautions must be taken with some vegetables (e.g., eggplants) that are particularly susceptible to post-cooking browning caused by the formation of dark complexes [130]. The problem may be reduced by adding chelating agents or antioxidants during blanching. Barbagallo et al. [131] used water-containing antioxidants (calcium ascorbate, 0.4%) for these blanching treatments. In other cases, cut eggplants are immersed in dry salt or brine. This may be useful to partially remove water and to reduce bitterness. By reducing O_2_ partial pressure immersion in brine may reduce browning. Zaro et al. [130] reported that a 1 h salting treatment may lead to extensive leakage of phenolic antioxidants (10–50%). As for other vegetables dips in ascorbic acid (or its less expensive isomer erythorbic acid) and citric acid are among the most common pretreatment used to minimize browning. Once the vegetables have softened, they are homogenizing/sieving to obtain the desired texture of the puree. Deaeration is also recommended to reduce air content in the product and to improve the heat treatment which is the next step; in this operation heat exchangers are the most extended equipment, but there is new non-thermal technologies, such as high hydrostatic pressure (HHP) increasingly used in pasteurization. Aseptic filling at ambient temperature or hot filling can be chosen according to the processing equipment. Blanching retained more phenolic, as it happens with freezing and canning. Conversely, stir-frying, boiling or steaming, reduced the leaves of these compounds in vegetables [127]. Kaur and Kaur [132] determined that blending and heat treatment to make vegetable puree, results in higher carotenoids, phenolics, and antioxidant capacity for peppers. Jez et al. [129] studied the HHP processing, defined as a mild technology, effect onto the carotenoids and other antioxidant compounds in tomato purees finding that carotenoids, as highly unsaturated compounds, are affected to chemical changes, like oxidation during food processing (lutein is strongly affected); meanwhile, lycopene bioavailability was enhanced during thermal processing. Since conventional thermal treatments can result in nutritional loss, the use of HHP technology, can improve the preservation and increasing the bioaccessibility of these compounds in juices and purees [133].

Most of the times, vegetables are industrial thermal processing or domestic cooking before being consumed and these treatments induce changes in nutritional composition and bioactive compounds both, in positive (many protective compounds are enhanced when vegetables are cooked because cooking facilitates the release of them that become easily available) and negative ways (thermal processing can cause degradation and, sometimes, leaching [134]. Texture and color are also greatly affected [135]. In general, thermal processing cause losses on nutritional and bioactive molecules, the loss of soluble compounds (sugars and soluble fiber) and affects color, hydration properties, and cell integrity [133,136]. As it was said before, some processes have a beneficial effect, like the release of soluble compounds, as polyphenols, that become available due to processing can disruption of the food matrix, increasing the bioaccessibility of many bioactive compounds [128,134]. The occurrence of these reactions depends directly on several factors such as the concentration of oxygen, metals, enzymes, unsaturated lipids, pro-oxidant and antioxidant compounds, exposure to light, severity of heat treatment, packaging, storage conditions, among other factors [137]. Time–temperature is an important factor to be considered in all types of preparations, although other aspects of the food product and method of preparation also need to be considered. For example, steaming has been revealed as a good method (better than boiling) to reduce the loss of fat-soluble nutrients during preparation of vegetables. Low losses in the concentration of β-carotene (close to 15%) were noted when steaming pumpkins for 40 min [138].

Special attention has received the genus Brassica (cabbage, kale, cauliflower, broccoli, brussels sprouts, etc.) which contains several non-nutrient phytochemicals, including isothiocyanates and glucosinolates [139,140,141,142]. The research of Pellegrini et al. [128] related to the effect of processing on phytochemicals of Brassica vegetables, demonstrated that all of the food processing (cooked by boiling, frying, and microwaving) affected the antioxidant properties and total polyphenols in cauliflower and other vegetables; meanwhile, Fouad and Rehab [143] showed that blanching and boiling caused losses of protein, mineral, and bioactive compounds and steam processing, stir-frying, and microwaving resulted in the retention of nutrients and phytochemicals contents. Boiling and microwaving generally resulted in an increment of antioxidant capacity in eggplant [128]. However, boiling of orange fleshed sweet potato (tuber) reduces the carotenoid content to varying degrees [144]. Dutta et al. [145] reported that, in the temperature range of 60–100°C for 0–2 h, the degradation of β-carotene in pumpkin puree during heating followed first-order reaction kinetics. While after 2 h at 60 °C, the concentrations of carotenoids were close to 14 μg/g, after the same period of time at 100 °C, the concentrations decreased to slightly over 4 μg/g. Gaur et al. [146] reported color changes from bright green to olive brown during the thermal processing (80–140 °C) of green leafy vegetables, attributed by conversion of chlorophylls to pheophytins due to the loss of central magnesium ion. These studies also revealed that both pigment and visual color degradation during thermal processing follows the first order reaction kinetics, being these effects stronger at acid pH. Martinez et al. [142] found that the effect of treatments varies considerably depending on the conditions and for each bioactive component.

In this way, Provesi et al. [137] compared the effect of cooking versus commercial sterilization on the stability of carotenoids in pumpkin puree. Contrary to the expected outcome, the major losses occurred during the cooking stage, despite the temperature being lower than that used in commercial sterilization. The reason for this is that cooking in water is carried out in open systems, with direct contact with oxygen causing lixiviation of nutrients, while in commercial sterilization the purees are already bottled, in closed systems, with a lower concentration of dissolved oxygen. Moreover, they reported higher losses in xanthophylls (>80%) than in carotenes (<25%), which was attributed to the presence or not of some oxygen groups in their structure; xanthophylls have one or more oxygen groups, and therefore lower oxidation stability. Conversely, carotenes, such as α-carotene and β-carotene, which are compounds that have no oxygen in their structure, obtained retention index relatively higher after processing.

During storage of these products, factors such as temperature, light exposure, and contact with oxygen may also cause loss of nutrients and bioactive compounds. The type of product, its conservation, and the packaging used play an important role in this context.

In the case of legumes (pulses), prior to consumption, their process (sometimes soaked, cooked in boiling water, steamed or toasted) varies their nutritional value too [147,148]. The four cooking methods (traditional or boiling, pressure, microwave, and slow) showed differences among legumes (faba beans, lentils, and peas) based on their phenolic content and/or antioxidant activity. After the processing, there were still high amounts of antioxidant compounds in cooked legumes [148]. In lentils, roasting process affects positively the physicochemical properties and the flavonoids; nevertheless, other bioactive compounds, such as, phenolics, carotenoids contents, were significantly reduced [110]. The results of the work of Arévalo et al. [147] showed that steam and toast processes retain protein, fiber, fat, and total phenolic contents as high levels of antioxidant capacity for chickpeas.

Grundya et al. [149] reported that the processing of amaranth (pseudocereals) caused changes on the digestibility of lipids and proteins.

The cereals refer to wheat, maize, rice, oat, rye, barley, millet, sorghum, etc. The whole cereal consists of whole parts of cereals (endosperm, germ, and bran) in the same percentage as in the anatomical forms. The whole cereal contains higher content of bioactive nutrients than refined ones. The processing induces changes in nutritional composition and in starch structure, producing starch gelatinization and retrogradation [150]. Rice as an important source of minerals and vitamin B, represents one of the most important staple foods. Washing and soaking are precooking processes of rice. Cooking, high-pressure or and microwave, are frequently used to cook the rice, result in loss of minerals and vitamins [148].

## 4. Health Benefits of the Intake of Vegetable Soups and Creams

The vegetable soups and creams can be considered healthy food due to their general composition and to their content in bioactive compounds (as it has been discussed in point 2), both attributed to their main ingredients (vegetables, legumes, cereals, etc., alone or combined). In addition, they are cheap and easy to preserve and prepare at home, so in consequence they are very useful in the modern life rhythms that modify the habits of current consumption [108,151].

However, the beneficial effects associated to the consumption of vegetable soups and creams are due not only to their nutritional properties, but also to other actions such us their promotion on the body’ hydration and to their satiating properties.

### 4.1. Nutritional Properties

In general, vegetable soups are valuable sources of mineral, vitamins, fiber, and other nutrients which are usually in short supply in daily diets. Buren et al. [152] studied the nutritional composition of dry vegetable soups (tomato, onion, pumpkin, and lentils) and compared these with published data on home-made and other soups. They found that the nutritional value of dry vegetable soups does not seem different from that of home-made soups. The dried vegetable soups can contain between 1 and 2.5 portion equivalents of vegetables, thereby contributing to 30–80% of the daily vegetable recommendation of 240 g/day [153]. Furthermore, they reported that most of the fiber and nutrients from the vegetable ingredients remain present in the dried soups and that the nutrient density of dried soups is like that of home-made and other vegetable soups. In view of these results the concluded that dry vegetable soups can be considered a suitable source of vegetables and nutrients and may deliver a significant part of recommended daily nutrient and vegetable intake. Furthermore, the nutritional quality of these soups can be easily improved (they even can be made suitable for specific food disorders) changing the amount or the type of vegetable used, adding new ingredients to enrich specific nutrients group, etc.

As has been commented, vegetable soups and creams are important sources of total dietary fiber. The adequate intake of dietary fiber in adults is of 25 g/day to be adequate for normal laxation. There is evidence of diets rich in fiber-containing foods (>25 g per day) reduced risk of coronary heart disease and type 2 diabetes and improved weight maintenance [154]. The main sources of soluble and insoluble dietary fiber are whole grain cereals, pulses, fruits, and vegetables and potatoes. Furthermore, fruits, vegetables, teas, and other herbal extracts usually used as ingredients in soup formulation contain a large group of compounds that could help maintain the body’s health and well-being. These are plant secondary metabolites, which, when consumed with a proper diet, can reduce the risk of age-related chronic diseases, protecting the body against damage and providing mechanisms to reduce free radicals induced by oxidative stress [135]. These secondary metabolites include phenolic acids, flavonoids, and glucosinolate derivatives, including isothiocyanates, terpenes, and low-molecular weight sulfur compounds, carotenes, among others, as has been previously discussed in point 2. Several examples can be found in the scientific literature about the enhancement of the phytonutrient content of soups using a composite of vegetables. Manhivi et al. [155] formulated an antioxidant rich and reduced salt dried vegetable soup made form a cocktail mix of leafy vegetables combined with pumpkin and sweet potato flour as the major starch sources. This resulted in a soup containing a variety of phenolic compounds (67.74 mg/kg, mainly catechin), chlorophylls (15.40 mg/kg, mainly chlorophyll A) and carotenoids (6.76 mg/100 g, mainly β-carotene) and with a significant increase in the free radical scavenging activity (DPPH IC50: 5.04 mg/mL). In addition, this soup contained a relatively lower amount of carbohydrates and sodium, while high in fiber. The nutritional quality could be improved by introducing protein, minerals, and vitamin sources from plant origin that are suitable for all types of people. Farzana et al. [7] enhanced the nutritional quality of a vegetable soup using soybean, mushroom, and moringa leaf (*Moringa oleifera*). It is also important to note that the resulted soup was specifically high in protein, ash, fiber, vitamin D, vitamin C, sodium, potassium, manganese, zinc, and iron and low in fat and energy value which make it an appropriate choice for the fulfillment of nutritional demand of the population. It has been studied the carotenoid content in different soups and cream showing higher levels those whose main ingredients were tomatoes and carrots. While the “red pepper and tomato soup” and “carrot and coriander soup” presented a carotenoid content of 3.18 µg/100 g and 2.58 µg/100 g, respectively; the “cream of leek soup” and “cream of mushroom soup” showed lower values between 0.07 µg/100 g and 0 µg/100 g, respectively [156].

At this point, a mention about some negative aspects (for both nutrition and health) that could be related with the consumption of this type of vegetable foods must also be made. It is regarding to food contaminants, and food intolerances. Food contaminants have emerged as a serious concern with potential health hazards in their wake. Majority of the food contamination occurs through naturally occurring toxins in specific vegetables (goitrogens glucosinolates in *Brassica* spp., furocoumarins in carrots, oxalic acid in spinach, and phytic acid in legumes and grains, among others) [157] and environmental pollutants (which are widespread and persistent contaminants, such as heavy metals from soil, pesticides residues as plant protection, or polychlorinated biphenyls in polluted areas) [158,159] or during the processing, packaging, preparing, storage, and transportation of food. Although the governments have taken adequate steps to minimize the individual exposure to food contaminants, there are still measures that need to be taken to reduce the health risks and diseases that come with the chemical food contamination. In this way, food industries must accept the need to be more honest and upfront in producing safe commercial food products as well as protecting the public from food contamination. The consumption of vegetable soups could be related to some food intolerances [160]: fermentable oligo-di-mono-saccharides and polyols (FODMAPs, present in wheat, onion, garlic, and legumes), gluten related (cereals), histamine (eggplant, spinach, and tomatoes) and lactose intolerances (if dairy products are used for providing creamy). In these cases, the information specified on the label should prevent the presence of such compounds to avoid health implications.

Table 1 shows the nutritional quality of some commercial vegetable soups, and of reformulated soups trying to increase their nutritional quality [145,146,149,161,162,163,164]. 

The recommendations of health organizations [154,165] are an intake of at least 400 g (that is, five servings) of fruits and vegetables a day to reduce the risk of developing non-communicable diseases (protective against cardiovascular disease and certain cancers) and to help to ensure a sufficient daily intake of dietary fiber. Despite the nutritious benefits and the epidemiological evidence of consuming fruits and vegetables, a significant majority of consumers still do not reach the recommendations of health organizations so the vegetables soups and creams could be an alternative to consumption, for its ease of preparation and preservation [166].

### 4.2. To Promote Body Hydration

The vegetable soups and creams are water-rich foods, and they may be helpful both for hydration and for dietary quality. The main ingredient is the water (between 60 and 95 % in soups and 60–77% in cream) [167], so they may help to maintain proper hydration and contributes to its low-calorie intake. Water fulfils a number of vital functions in the body, as it is the main component of the body fluid (saliva, synovial fluids, urine, blood, vitreous humor, and tears, etc.), as well as all the biochemical event in the cells are in water [168,169].

The recommended water intakes range from 2.5 to 3.7 L per day for adult men and from 2.0 to 2.7 L per day for adult women [170]. These water requirements are provided from drinking water, beverages and the water contained in the dietary foods. Foods provide about 20% on intake of water, but this could depend on the kind of food chosen. The inclusion of the vegetables cream and soups in the daily diet, as water-rich foods, may be beneficial for maintain the hydration or the balance of water on the body [168].

### 4.3. Satiating Properties

In order to control their diet and body weight, the consumers are demanding high satiety products. The dietary fiber stand out among the other food ingredients that could have effects on satiety and depending on different factor, as the fiber type (soluble and insoluble), and its ability to bulk foods (viscosity, gel in the stomach, and ferment in the gut) [171]. These authors reported that food with high content in protein and fiber could have positive effects on appetite control. Moreover, rich-fiber meals can promote satiety earlier and it is associated with a lower calorie intake [172]. Not only the fiber, but also the antioxidants compounds, together with the large amount of water in these products could contribute to the feeling of satiety and avoid a greater intake of food, appetite control, and contribute to healthy and the beneficial effects.

Some studies have concluded than there were an inverse association between soup consumption and body weight in population from different countries (France, Portugal, Japan, and US) [173,174,175,176]. This effect was attributed to low energy density and large content of liquid and the satiating effects of soup recommending, an increase in the intake of soup consumption for its association with the reduced risk of overweight or obesity [176]. Flood and Rolls [177] suggested that the if the soup were eating before a meal could increase the satiety and reduced intake, due to enhanced gastric distension and a decreased rate of gastric emptying.

## 5. New Trends in the Development of Vegetable Soups and Creams

The consumer has become the center of the innovation chain and they are now driving the changes that occur in the food industry in terms of new product development. In view of that, one of the challenges the food industry faces is balancing the provision of what consumers want to buy with what is important from a public health perspective, recognizing that there is often a tension between consumers’ desire for choice and choice editing by manufacturers, retailers, and the food service sector [178]. One of the most interesting and novel proxies of healthy diets are food, namely, produce availability [179] Interesting research is suggesting that the easier it is to purchase healthful foods, the easier it is to follow appropriate diets [180,181].

Reformulation of food and drink products would be of prime nutrition policy importance. It justifies “public–private partnerships” at which agreements concerning product formulation are made. Reformulation that reduces the amount of fat, sugar, and salt, or that increases the amount of dietary fiber, vitamins, minerals, or other bioactive compounds, will improve the nutrient profile of processed products [182]. Therefore, it would result in healthier food supplies and dietary patterns and help to control and prevent obesity and chronic non-communicable diseases, as specified at the General Assembly of the United Nations [183]. Highlighting the relevance of “healthy” in food purchase; nowadays, there are other aspects with a high impact in the decision of food purchase and consumption.

According to the annual global food trends report of Mintel (Market Intelligence Agency) [184], the concepts around which food trends revolve are trust in the traditional, the power of plants, the time is crucial (convenience foods), zero residue, and cult of “healthy”. Around all these concepts, the trends with a high influence in the processed vegetables sector are as follows.

### 5.1. Healthier Nutrition

The consumer has a growing concern in his correct nutrition and health. Overweight and obesity, as well as their related noncommunicable diseases (cardiovascular diseases, diabetes, musculoskeletal disorders, and some cancers) which have a very high and worrying incidence in our current society are largely preventable. The World Health Organization launched the “WHO Global Strategy on diet, physical activity, and health” [165] describing the actions needed to support healthy diets and regular physical activity. These actions involve decisions not only at individual level (individual responsibility to follow a healthy lifestyle) but also the food industry can play a significant role in promoting healthy diets (reducing the fat, sugar, and salt content of processed foods, ensuring that healthy and nutritious choices are available and affordable to all consumers, restricting marketing of unhealthy foods and ensuring the availability of healthy food choices) [185]. To reinforce these actions, world and European authorities tend to be more demanding every day increasing restrictions for unhealthy ingredients in foods. The range of innovative solutions in response to these restrictions includes:Substitution of animal protein by vegetable proteins: Overwhelming evidence shows that overconsumption of meat is bad not only for human health but also for environmental health and that moving towards a more plant-based diet is more sustainable [186]. This change is urgently needed to ensure One Health objectives (United Nations Sustainable Development Goals) for people and the planet are achieved [187]. The innovation opportunities in this sector include the substitution of thickened agents from animal origin (milk, cream, butter, meat, or fish broth, etc.) by others from vegetable origin such as certain legumes [151,152,188,189], ancient cereals [188,190], pseudocereals [191,192], seaweeds [193], and others.Salt reduction: In this aspect, it should not be forgotten that canned soup is one of the leading processed food categories containing high quantities of sodium. Although it has been suggested that intake of sodium rich foods in excess of immediate bodily needs is a preventative mechanism serving to embed sodium sources in memory and thus ward off hyponatremic challenge, nowadays much of the sodium that individuals and manufacturers place in foods is present because people “like” the taste of sodium rich food better than the taste of the same food without sodium [194,195]. WHO reported that high sodium consumption (>2 g/day, equivalent to 5 g salt/day) and insufficient potassium intake (less than 3.5 g/day) contribute to high blood pressure and increase the risk of heart disease and stroke. Most people consume too much salt—on average 9–12 g per day, or around twice the recommended maximum level of intake. In view of the severity of excess salt intake with foods the WHO Member States have agreed to reduce the global population’s intake of salt by a relative 30% by 2025. To achieve this goal, one of the proposed actions is that authorities have to work with the private sector to improve the availability and accessibility of low-salt products [196]. Several authors have studied the influence of salt taste threshold on acceptability and purchase intent of reformulated reduced sodium vegetable soups and the results highlighted the importance of both taste characteristics and the inclusion of the consumer in the reformulation process of reduced salt foods [197,198]. The study from Mitchell et al. [198] also showed that salt reductions of up to 48% can be achieved in commercial vegetable soup samples without affecting consumers liking for the meal.Sugar reduction: The sugar content in commercial vegetable soups can result surprising. For example, Gallager [199] reported amount of sugar up to 20 g per tube in commercial root and tomato vegetable soups. The WHO recommends people take no more than 24 g a day. Traditionally the addition of sugar to soups (mainly as isoglucose) has been made with the goal of making the soup tastier, reducing acid taste, or increasing creamy sensation, but it is not necessary if the balance of ingredients is correct [200]. In this sense, most of the new soup formulations are not added sugars.Fiber addition: Dietary fiber, which is not easily hydrolyzed by the human digestive system, reduces the risk of cardiovascular diseases, obesity, diabetes, and gastrointestinal disorders. It mainly consists of lignin and plant polysaccharides, and their physiological effect is determined by physical and chemical properties, such as the polysaccharide degree of polymerization and crosslinking, cell wall integrity, and particle size. Although vegetables soups are considered good sources of dietary fiber (depending on the main vegetable used in their formulation) [146], the deficit in the amount of dietary fiber in our current diet has led to an increase in the supply of fiber-enriched foods. Apart from the nutritional purpose, fiber can be used in vegetable soups for technological purposes as bulking or gelling agent or fat substitute [201]. Some of the ingredients used for this enrichment are quinoa grains (13–15% dietary fiber content, mainly insoluble dietary fiber) [202,203], legumes (7–10% dietary fiber content depending on the type) [189], and fiber extracts obtained from coproducts of vegetable-food processing [204,205,206]. These coproducts represent a major disposal problem for the industry and its transformation into value-added products, as fibers are, may contribute to diminish the problem and to recover valuable biomass and nutrients [204,205,206]. The use of fibers from new origins that are currently not fully exploited and the possibility of modifying the fibers by chemical, enzymatic and/or physical treatments will probably widen the fields of application for dietary fibers [207,208].Fat reduction: Consumer concerns about excess fat consumption of some types of lipids have led to the development by the food industry of a new category of foods to meet this demand. The development of reduced-fat foods with the same desirable attributes as the corresponding full-fat foods has created a distinct challenge to food manufacturers. In view to achieve this nutritional item, different vegetable proteins and starches have been applied as fat substitutes in soups. Several authors have reported that the water binding enhancement by whey proteins is important in reduced-fat and fat-free formulations to retain the increased water in the formula, and to replace fat and maintain texture and yield [209]. Whey proteins contribute to increased opacity and help reduced-fat soups and sauces retain a good visual appeal, with the creamy appearance of regular products. Whey proteins provide fat-like functions of lubricity and significantly improve the mouthfeel of low-fat soups and sauces. Emulsification properties of whey proteins aid in efficient dispersion of fat by forming interfacial membranes around oil globules that prevent creaming, coalescence, and oiling off [210,211]. In addition, whey proteins contribute to increased shelf-life in terms of product appearance and consumer acceptability. Interesting results have also been obtained using cross-linking tapioca starch (CLTS) as fat substitute in soups. Variations of swelling power, solubility, pasting, gelatinization, and rheological properties of the CLTS were found. Thermogravimetric analysis exhibited higher thermal stability for the CLTS granules compared to the native one. Among the samples, soup containing the 1.0%-CLTS exhibited the strongest gel characteristic and the greatest shear resistant properties. The 1.0%-CLTS improved the textural properties and sensory quality of soups [212]. Cox et al. [213] tried to characterize the changes in the drivers of liking when sodium and fat levels were varied in a model retorted soup system. They reported that formulation modifications that would decrease intensities of attributes that characterize lower fat and sodium soups, such as sour (taste and aftertaste), grainy (texture), and darkness (appearance), will aid in higher consumer acceptance of these soups.

### 5.2. High Added Value

A recent study about European consumer trends showed that 56% of respondents consider that processed vegetable foods are a good way to try new recipes and formulations and more than 20% are willing to pay more for hedonic food products made with premium and gourmet ingredients [214]. They understand that this type of food contains only the highest quality fresh ingredients and its preparation is supervised by highly trained chefs and so the consumption of the final product will result in a top gastronomy experience unique and retrievable. In this case, both the social challenges and the innovation opportunities are related to the use of premium raw materials or with official quality brands (European, National, or Regional) indicating the origin of this ingredient (Geographical Indications) and its production way.

On the other hand, vegetable soups with non-traditional ingredients that provide new and exotic flavors are also a good option to increase the added value of these products. For example, some of the main multinational food companies are exploring this market niche launching exotic and ethnic soups: “Thai-style” soup with coconut cream, lemongrass, and ginger; “Mexican-style” soup with tortilla, black beans, corn, and chili pepper, etc.

### 5.3. More Natural (Clean Label, Organic Ingredients, Less Invasive Production Processes, Coproducts Valorization, etc.)

Although more than 70% of European buyers think that vegetable ready meals are a good option when you have to eat alone or you are very busy for cooking, 40% consider that ready meals are too processed and unnatural and so, if there are other options of convenient and natural food, they select these to the detriment of the ready meals. The innovation opportunities to overcome this problem go through removing additives (no colorants or preservatives), less invasive production processes, valorization of coproducts from agro-industries, packaging systems environmentally friendly that allow water vapor cooking, and sensorial quality improvement.

Cornelia and Christianti [215] studied the valorization of the avocado (*Persea americana* Mill.) seeds, that often are seen as waste and underutilized resources, especially in the food industry, as source of starch, modifying its structure using the cross-linking method, to improve the viscosity stability in the cream soup. The result showed that cream soup using modified starch has better viscosity stability than native starch and commercial cream soup after five hours storage.

### 5.4. Shelf-Life Innovations

Drying process has a great impact on the quality and shelf-life of dehydrated soups so, new drying procedures are been implemented to improve their final quality. The ability to predict the moisture content during storage under a variety of conditions is very important for reducing the cost and the cycle time of product development. In this way, Singh and Prasad [216] studied the moisture sorption characteristics of rice-based instant soup to improve the drying process toward more economic and energy reduced treatments without sensorial properties detriment. Wang et al. [217] studied the effect of the use of a microwave freeze dryer upon the sensory quality of instant vegetable soups reported a significant influence in both, the total drying time and sensory quality of the final product.

Other option to increase the shelf-life of soups is by freezing. Currently it is possible to found different types of frozen vegetable soups presented in a single-serving bowl ready to go from microwave to table. The freezing process significantly increases their shelf-life and allow (after thawed) to offer a product with a homemade flavor and freshness that canned versions just do not have [218]. Packaging innovations (materials and types) can also contribute to increase the shelf-life of the soups. It can be said that the soup sector has pioneered its transformation from steel can to aseptic plastic and cartons with different shapes and sizes (health and wellness coupled with convenience). Shelf-stable re-closeable packaging formats (designed for conventional, full-size family formats) are decreasing in favor of individual format (“on go”) due to the interest of millennial consumers [219].

### 5.5. Products Aimed at Specific Population Groups

Although creams and soups can be consumed for nutritive benefits by the entire population, the selection of ingredients, their texture and appearance could be managed to aim at specific populations groups.

6.Adults can reduce their energy intake from the entire meal by consuming low-energy dense foods before the main course as a first course, increasing their satiety or even for diner due to its easy digestion which is positive for comforting and not unbalanced sleep. Regarding that, The Spanish Society of Communitarian Nutrition recommends the introduction of soups and creams at dinners [220].7.Children’s vegetable consumption is generally far below the recommended standards, and effective strategies are required to support additional consumption [221]; attractive and nutritive soups and creams could increase their vegetable consumption.8.Elderly people are encouraged to consume soups and creams four or more times a week because helps to achieve the essential fluid intake to meet daily requirements due to their high water content. In addition, their high vitamins and minerals content contributes to achieve others daily nutritional requirements, mainly in elderly [169].9.Celiacs. Celiac disease is a major public health problem worldwide. Approximately 3 million people in Europe and another 3 million people in the United States are estimated to be affected by celiac disease [222]. Traditionally semi-prepared soups and creams are made of wheat (or another gluten-containing cereal) and texturizing agents to contribute to their creamy taste and texture because it has been reported that the absence of gluten leads to a low viscosity [223]. Several ingredients have been tested in order to make these food products suitable for celiacs, being quinoa flour, which showed the better results. Quinoa has a great ability to form emulsions, a high water holding capacity and high solubility. Its high amylopectin content contributes to its ability to reach the gelatinization temperature in less time than corn and in a similar time as wheat with lower retrogradation. This represents a good gel stability [224], which is difficult to achieve when wheat flour is used to produce semi-prepared soups and creams. Quinoa flour is an excellent ingredient for soups and creams not only for these technological properties but also for its nutritional value (50–60% carbohydrates, 7–10% dietetic fiber, 12–23% protein content with great amino acid balance, and vitamins [225].10.Patients whose intake of solids is considerably reduced due to several pathological reasons. In this case, the formulation must be optimized in view of compensating the small food volumes with the highest nutrients and energy without misleading aspects such as an adequate texture that allows its correct deglutition [169].

Depending on the type of soup, claims related to health, organic production, environmental protection or specific population groups have been used to improve their marketing. Slogan as “clean ingredients”, “natural ingredients”, “no-added sugars”, “low-sodium or light-in-sodium”, “organic soup”, “non-GMO ingredients”, “full of fiber”, “gluten-free”, “low-fat”, and “low-calories” among others, are been used as quality differentiator respect to other commercial soups.

In conclusion, vegetable soups can be considered a suitable source of vegetables and nutrients which can be easily reformulated to fulfill nutrient requirements for specific population groups or with regards to health benefits, without misleading their gastronomic value. This nutritional value depends on the vegetables used for their production, most of which are recognized sources of dietary fiber and bioactive compounds (mainly polyphenols) with antioxidant and healthy properties. Research and innovation efforts have been applied to improve their processing in view of that most of these beneficial compounds could be retained in the final product, and also to offer foods that meet the necessities of today’s society: trust in the traditional, the power of plants, the time is crucial (convenience foods), zero residue, and wellbeing.

## Figures and Tables

**Figure 1 plants-09-01769-f001:**
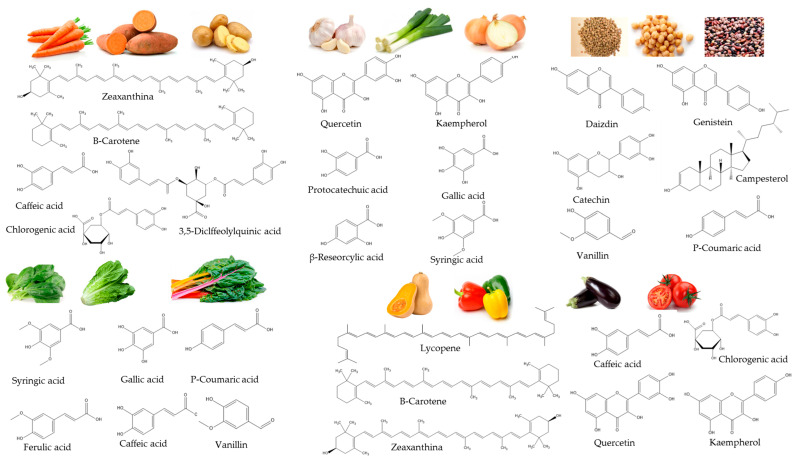
Main bioactive compounds present in the ingredients frequently used in the preparation of vegetable soups.

**Figure 2 plants-09-01769-f002:**
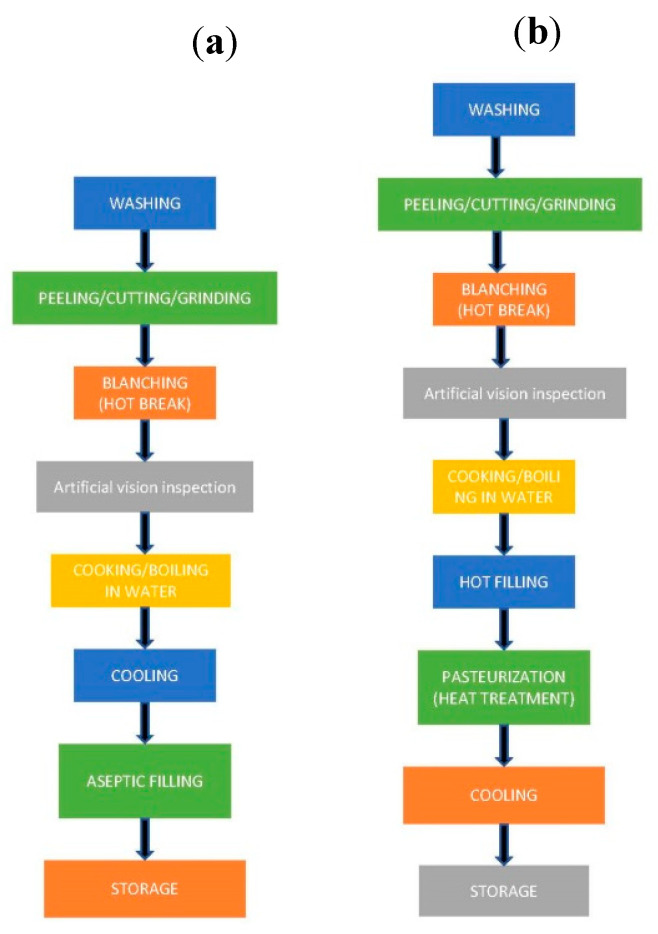
Flow-chart of processing vegetable soups packed using aseptic filling (**a**) or hot filling (**b**).

**Table 1 plants-09-01769-t001:** Nutrient content of commercial (C) and reformulated (R) vegetable soups (per 250 mL serving).

Type of Soup ^1^	Energy	Proteins		Carbohydrates		Fiber		Fats		Sodium		Reference
	(Kcal)	g	%DRI ^2^	g	%DRI	g	%DRI	g	%DRI	mg	%DRI	
Tomato S. (C)	82	1.9	4.8	16.3	32.6	nd	-	1.0	3.7	nd	-	[151]
Tomato S. (C)	128	1.5	3.8	16.5	33	traces	-	6.3	23.3	1800	90	[151]
Tomato S. (C)	110	-	-	-	-	2.9	11.6	-	-	856	42.8	[146]
Tomato S. (R)	83	2.3	5.8	18	36	1	4	2	7.4	500	25	[151]
Onion S. (C)	58	-	-	-	-	5.2	20.8	-	-	843	42.2	[146]
Pumpkin S. (C)	98	-	-	-	-	2.8	11.2	-	-	891	44.6	[146]
Lentil S. (C)	155	-	-	-	-	11	44	-	-	825	41.3	[146]
M. Veget. S. (C)	90	-	-	-	-	4.3	17.2	-	-	713	35.7	[146]
M. Veget. S. (C)	44	1.0	2.5	8.2	16.4	nd	-	0.8	3.0	840	42	[151]
M. Veget. S. (C)	110	0.9	2.3	8.8	17.6	4.6	18.4	8	29.7	1686	84.3	[151]
M. Veget. S. (C)	41	0.6	1.5	8.1	16.2	2.4	9.6	0	0	90	4.5	[152]
M. Veget. S. (C)	41	1.1	2.8	9.4	18.8	1.1	4.4	0.9	3.3	355	17.8	[149]
M. Veget. S. (R)	175	3.9	9.8	21	42	9.2	36.8	8.3	30.7	325	17.8	[151]
M. Veget. S. (R)	42	1.0	2.5	6.5	13	2.4	9.6	0.8	3.0	168	8.4	[149]
Mushroom S. (C)	59	2.3	5.8	9.9	19.8	nd	-	1.1	4.1	720	36	[151]
Mushroom S. (C)	82	1.8	4.5	9.7	19.4	0.5	2	4	15.8	875	43.8	[151]
Mushroom S. (R)	71	2.0	5.0	13	26	0.8	3.2	1.3	4.8	400	20	[151]
Veget. C. (C)	97.5	2.5	6.3	12.75	25.5	3	12	3.75	13.9	500	25	[153]
Vichyssoise (C)	145	1.5	3.8	12.75	25.5	0.5	2	9.25	34.3	480	24	[153]
Pumpkin C. (C)	92.5	2.0	5.0	10	20	2.25	9	5	18.5	760	38	[153]
Asparagus C. (C)	67.5	1.75	4.4	9.25	18.5	1.25	5	2.25	8.3	690	34.5	[153]
Veget. C. (C)	62.5	1.75	4.4	11	22	3	12	0	0	750	37.5	[153]
Med.Veget. S. (C)	115	2.25	5.7	9.75	19.5	3	12	6.5	24.1	650	32.5	[153]
Mushroom C. (C)	52.5	1.5	3.8	9.75	19.5	0		1.25	4.6	830	41.5	[153]
M. Veget. S. (C)	92.8	1.8	4.5	19.86	39.7	0.41	1.6	0.70	2.3	-	-	[145]
M. Veget. S. + lentil (R)	93.8	3.74	9.4	17.99	35.98	0.40	1.6	0.78	2.9	-	-	[145]
M. Veget. S. + green peal (R)	93.6	3.72	9.4	17.82	35.6	1.05	4.2	0.83	3.1	-	-	[145]
M. Veget. S. + chickpea (R)	96.2	3.38	8.5	17.82	35.6	0.45	1.8	1.28	4.7	.	-	[145]

^1^ S.= soup; (C)= commercial; (R)= reformulated; M. Veget.= mixed vegetables; C.= cream. ^2^ %DRI=Dietary reference intake [164]: for protein, this value was estimated for 60 kg body weight

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
