# Peer review of "Vegetable Soups and Creams: Raw Materials, Processing, Health Benefits, and Innovation Trends"

_plants, 2020, doi:10.3390/plants9121769_

Round 1
Reviewer 1 Report
This manuscript is not deserved for its publication in Plants in the present form. Some points should be improved and explained by the authors.
Remarks:
Page2, lines 119-122: The authors listing the main phenolic compounds present in peppers should carefully determine - Does it concern the seeds or the pulp of pepper? Authors giving the values and units of these compounds should also provide information - What does "Kg" refer to? In addition, are the values reported for ferulic acid really within the range 2661-404 µg /Kg?
Page 3, line 119: The authors reported that paprika is a source of fatty acids. Please list them.
The authors sometimes write the expression - "Vitamin E" in uppercase and sometimes in lowercase. Please unify it.
Page 4, line 149: Is chlorogenic acid 5-O-caffeoylquinic acid? Maybe it is isomer of chlorogenic acid? See page 7, line 293 of this manuscript.
Page 4, line 162: What does the symbol C3G mean?
Page 4, line 163: The antioxidant properties of the peppers were measured by ABTS and were expressed in µg. Is this the correct unit?
Page 6, line 247: The phrase "vitexin-2-O-rhamnboside" should be changed to "vitexin-2-O-rhamnoside".
Page 6, line 248: Page 6, line 248: Please specify which compounds contribute to the antioxidant activity of Swiss chard?
Page 7, line 330: What is this peonidin-3-soph-5-glucose?
Page 9, line 422: The expression "cupper" should be changed to "copper".
Figure 1: The expression "β-caroteno" should be changed to "β-carotene".
The authors described the raw material used in the production of soups and creams with reference to their ingredients, especially phenolic compounds, minerals and vitamins. However, in the subsection "Processing effects on the main components of raw materials" there is no direct reference to these raw materials, e.g. eggplant, pumpkin or Swiss chard.
References should be checked carefully. Please check and correct at each point of references whether the Latin plant names are in italics.
Author Response
We thank all your comments and suggestions that allow us to clarify the message of our paper.
The paper has been carefully revised and language and grammatical errors have been corrected.
I am going to answer all your comments point by point. Your comments are in black and our answers in blue color.
Page2, lines 119-122: The authors listing the main phenolic compounds present in peppers should carefully determine - Does it concern the seeds or the pulp of pepper? Authors giving the values and units of these compounds should also provide information - What does "Kg" refer to? In addition, are the values reported for ferulic acid really within the range 2661-404 µg /Kg?
- Several data about phenolic compound found in pepper have been added. Thus the samples analyzed were the pulp after remove the placenta and seeds (this information has been incorporated into the text).
- Kg refers to sample fresh weight (this information has been incorporated into the text)
- The ferulic acid content ranged from 404 to 2661 µg /Kg of fresh weight
Page 3, line 119: The authors reported that paprika is a source of fatty acids. Please list them.
- The main fatty acids present in pepper have been incorporated into the text
The authors sometimes write the expression - "Vitamin E" in uppercase and sometimes in lowercase. Please unify it.
- This has been revised and one error in line 145 has been corrected
Page 4, line 149: Is chlorogenic acid 5-O-caffeoylquinic acid? Maybe it is isomer of chlorogenic acid? See page 7, line 293 of this manuscript.
- Thanks for your correction. 5-O-caffeoylquinic acid has been change by 3-O-caffeoylquinic acid, that is the correct nomenclature of chlorogenic acid
Page 4, line 162: What does the symbol C3G mean?
- C3G is the abbreviation of Cyanidin-3-glucoside. C3G has been replaced by Cyanidin-3-glucoside
Page 4, line 163: The antioxidant properties of the peppers were measured by ABTS and were expressed in µg. Is this the correct unit?
- No, you are true; there is a mistake. The correct units are µg / mL. Now it has been corrected in the text
Page 6, line 247: The phrase "vitexin-2-O-rhamnboside" should be changed to "vitexin-2-O-rhamnoside".
- Thanks, your suggestion has been added
Page 6, line 248: Page 6, line 248: Please specify which compounds contribute to the antioxidant activity of Swiss chard?
- The compounds that contribute to the antioxidant activity of Swiss chard are phenolic acids, flavonoids and anthocyanins. This information has been incorporated into the text as you suggested
Page 7, line 330: What is this peonidin-3-soph-5-glucose?
- Peonidin-3-soph-5-glucose is the abbreviation of peonidin-3-sophoroside-5-glucose. Now, peonidin-3-soph-5-glucose has been replaced by peonidin-3-sophoroside-5-glucose in the text.
Page 9, line 422: The expression "cupper" should be changed to "copper".
- It was an error and now “cupper” has been changed by “copper”
Figure 1: The expression "β-caroteno" should be changed to "β-carotene".
- Thanks for the correction, “β-caroteno" has been changed by "β-carotene” in figure 1.
The authors described the raw material used in the production of soups and creams with reference to their ingredients, especially phenolic compounds, minerals and vitamins. However, in the subsection "Processing effects on the main components of raw materials" there is no direct reference to these raw materials, e.g. eggplant, pumpkin or Swiss chard.
- Yes, I agree with you. This section has been improved taken into account your suggestion, and more information about these effects on several raw materials has been added
References should be checked carefully. Please check and correct at each point of references whether the Latin plant names are in italics.
- All references has been checked and all binominal names are in italics

Reviewer 2 Report
The Manuscript “Vegetable soups and creams: Raw materials, processing, health benefits and innovation trends " by Juana Fernández-López, Carmen Botella-Martínez, Casilda Navarro-Rodríguez de Vera, María Estrella Sayas-Barberá, Manuel Viuda-Martos, Elena J. Sánchez-Zapata, Jose Angel Pérez-Álvarez,”focuses on the nutritional and healthy properties of vegetable soups and creams (depending on the raw materials used in their production) highlighting their content in bioactive compounds and their antioxidant properties”.
I believe that the Manuscript can be accepted after appropriate minor revisions, as detailed in the reviewer’s comments to the Authors.
Minor revisions:
Row 16: Change “weel- being” with wellbeing
Row 20: ..textures . space before dot
Row 405 change “galic acid” with gallic acid
Row 611 change “(as has been discussed in point 2) with (as it has been discussed in point 2)
The authors, row 94-98 “...summarize and discuss the recent advances about the composition and identification of bioactive compounds with healthy properties in the most common raw materials used in soup and creams formulation, the effect of processing on the main components of these raw materials and the innovation trends in the vegetable soups and creams sector. It is also aimed at exploring the health benefits associated to the consumption of vegetable soups and creams”.
Although the manuscript highlights the nutritional properties well, some important aspects such as the chemical contamination occurring during transportation and food processing/storage, that has emerged as an important issue with potential health hazards (heavy metals, PCB, micotoxins etc. ref. Adornetto A, Pagliara V, Renzo GD, Arcone R. Polychlorinated biphenyls impair dibutyryl cAMP-induced astrocytic differentiation in rat C6 glial cell line. FEBS Open Bio. 2013;3:459-466. doi:10.1016/j.fob.2013.10.008; Rather IA, Koh WY, Paek WK, Lim J. The Sources of Chemical Contaminants in Food and Their Health Implications. Front Pharmacol. 2017;8:830. doi:10.3389/fphar.2017.00830). In addition, ingredients that cause food intolerances (ref.: Tuck CJ, Biesiekierski JR, Schmid-Grendelmeier P, Pohl D. Food Intolerances. Nutrients. 2019;11(7):1684. doi:10.3390/nu11071684) such as lactose, are not discussed.
In light of these considerations, I would highly recommend including these topics in the review for its publication in the journal.
Author Response
We thank all your comments and suggestions that allow us to clarify the message of our paper.
The paper has been carefully revised and language and grammatical errors have been corrected.
I am going to answer all your comments point by point. Your comments are in black and our answers in blue color.
The Manuscript “Vegetable soups and creams: Raw materials, processing, health benefits and innovation trends " by Juana Fernández-López, Carmen Botella-Martínez, Casilda Navarro-Rodríguez de Vera, María Estrella Sayas-Barberá, Manuel Viuda-Martos, Elena J. Sánchez-Zapata, Jose Angel Pérez-Álvarez,”focuses on the nutritional and healthy properties of vegetable soups and creams (depending on the raw materials used in their production) highlighting their content in bioactive compounds and their antioxidant properties”.
I believe that the Manuscript can be accepted after appropriate minor revisions, as detailed in the reviewer’s comments to the Authors.
- Thanks a lot for your positive comments about our work. We will try to incorporate all your suggestions
Minor revisions:
Row 16: Change “weel- being” with wellbeing
- OK, it has been changed.
Row 20: ..textures . space before dot
- OK, it has been corrected.
Row 405 change “galic acid” with gallic acid
- OK, it has been changed.
Row 611 change “(as has been discussed in point 2) with (as it has been discussed in point 2)
- OK, it has been changed.
The authors, row 94-98 “...summarize and discuss the recent advances about the composition and identification of bioactive compounds with healthy properties in the most common raw materials used in soup and creams formulation, the effect of processing on the main components of these raw materials and the innovation trends in the vegetable soups and creams sector. It is also aimed at exploring the health benefits associated to the consumption of vegetable soups and creams”.
Although the manuscript highlights the nutritional properties well, some important aspects such as the chemical contamination occurring during transportation and food processing/storage, that has emerged as an important issue with potential health hazards (heavy metals, PCB, micotoxins etc. ref. Adornetto A, Pagliara V, Renzo GD, Arcone R. Polychlorinated biphenyls impair dibutyryl cAMP-induced astrocytic differentiation in rat C6 glial cell line. FEBS Open Bio. 2013;3:459-466. doi:10.1016/j.fob.2013.10.008; Rather IA, Koh WY, Paek WK, Lim J. The Sources of Chemical Contaminants in Food and Their Health Implications. Front Pharmacol. 2017;8:830. doi:10.3389/fphar.2017.00830). In addition, ingredients that cause food intolerances (ref.: Tuck CJ, Biesiekierski JR, Schmid-Grendelmeier P, Pohl D. Food Intolerances. Nutrients. 2019;11(7):1684. doi:10.3390/nu11071684) such as lactose, are not discussed.
In light of these considerations, I would highly recommend including these topics in the review for its publication in the journal.
- Thanks for your suggestion. A wide paragraph trying to resume all these aspects has been incorporated.

Reviewer 3 Report
The manuscript is a review on vegetable soups and creams and the influence of raw materials, processing treatments, health benefits and innovation trends.
Nevertheless, the manuscript needs to be improved in order to provide enough information to justify its importance and novelty.
Please use the section and restructure background to explain the importance and novelty of this manuscript and its objective.
Please conclude according to the main topics included in the paper.
In Table 1 should be included informations on % of each nutrient in daily intake (can be calculated).
Author Response
We thank all your comments and suggestions that allow us to clarify the message of our paper.
The paper has been carefully revised and language and grammatical errors have been corrected.
I am going to answer all your comments point by point. Your comments are in black and our answers in blue color.
The manuscript is a review on vegetable soups and creams and the influence of raw materials, processing treatments, health benefits and innovation trends.
Nevertheless, the manuscript needs to be improved in order to provide enough information to justify its importance and novelty.
Please use the section and restructure background to explain the importance and novelty of this manuscript and its objective.
- In our opinion the objective of the manuscript is clear but we have include a paragraph to highlight the importance and novelty of this review
“The concern about the relationship between diet and health, the growing demand for healthy ready-to-eat products, according to current social habits, the boom in diets based on veganism and the possibilities that vegetable soups offer to innovate and develop new products (ingredients, tastes, process, packaging etc.) are some of the reasons that have contributed to the current relevance of this type of foods. Considering that, in this review, we summarize and discuss recent advances about the composition and identification of bioactive compounds with healthy properties in the most common raw materials used in soup and creams formulation, the effect of processing on the main components of these raw materials and the innovation trends in the vegetable soups and creams sector. It is also aimed at exploring the health benefits associated to the consumption of vegetable soups and creams.”
Please conclude according to the main topics included in the paper.
- A conclusion paragraph has been included at the end of the paper.
In Table 1 should be included informations on % of each nutrient in daily intake (can be calculated).
- %Dietary reference intake haven been calculated and included in table 1

Round 2
Reviewer 1 Report
I have read the authors' responses and re-read the presented manuscript. I think the manuscript is suitable for publication in Plants.
Reviewer 3 Report
The manuscript has been improved. In this form can be published.